# HVCN1 but Not Potassium Channels Are Related to Mammalian Sperm Cryotolerance

**DOI:** 10.3390/ijms22041646

**Published:** 2021-02-06

**Authors:** Ariadna Delgado-Bermúdez, Yentel Mateo-Otero, Marc Llavanera, Sergi Bonet, Marc Yeste, Elisabeth Pinart

**Affiliations:** 1Biotechnology of Animal and Human Reproduction (TechnoSperm), Institute of Food and Agricultural Technology, University of Girona, E-17003 Girona, Spain; ariadna.delgado@udg.edu (A.D.-B.); yentel.mateo@udg.edu (Y.M.-O.); marc.llavanera@udg.edu (M.L.); sergi.bonet@udg.edu (S.B.); marc.yeste@udg.edu (M.Y.); 2Unit of Cell Biology, Department of Biology, Faculty of Sciences, University of Girona, E-17003 Girona, Spain

**Keywords:** cryopreservation, sperm, HVCN1 channels, SLO1 channels, potassium channels, pigs

## Abstract

Little data exist about the physiological role of ion channels during the freeze–thaw process in mammalian sperm. Herein, we determined the relevance of potassium channels, including SLO1, and of voltage-gated proton channels (HVCN1) during mammalian sperm cryopreservation, using the pig as a model and through the addition of specific blockers (TEA: tetraethyl ammonium chloride, PAX: paxilline or 2-GBI: 2-guanidino benzimidazole) to the cryoprotective media at either 15 °C or 5 °C. Sperm quality of the control and blocked samples was performed at 30- and 240-min post-thaw, by assessing sperm motility and kinematics, plasma and acrosome membrane integrity, membrane lipid disorder, intracellular calcium levels, mitochondrial membrane potential, and intracellular O_2_^−^⁻ and H_2_O_2_ levels. General blockade of K^+^ channels by TEA and specific blockade of SLO1 channels by PAX did not result in alterations in sperm quality after thawing as compared to control samples. In contrast, HVCN1-blocking with 2-GBI led to a significant decrease in post-thaw sperm quality as compared to the control, despite intracellular O_2_^−^⁻ and H_2_O_2_ levels in 2-GBI blocked samples being lower than in the control and in TEA- and PAX-blocked samples. We can thus conclude that HVCN1 channels are related to mammalian sperm cryotolerance and have an essential role during cryopreservation. In contrast, potassium channels do not seem to play such an instrumental role.

## 1. Introduction

Ion channels are essential for sperm physiology during spermatogenesis, epididymal maturation, and storage, transit throughout the female reproductive tract and interaction with the oocyte [1]. Since the first identification of cation sperm channels (CatSper) in the plasma membrane of mouse spermatozoa in 2001 [2], several studies have investigated different ion channels and their relevance for sperm function (reviewed in [3]). These studies were usually performed in mouse and human sperm and focused on the physiological role of these ion channels during sperm capacitation. In contrast, little data exist about the types of ion channels present in the sperm of other mammalian species, as well as on their functional significance in sperm differentiation, maturation and adaptation to the surrounding medium, and their relationship with sperm cryotolerance.

Cryopreservation is the most efficient method for long-term storage of sperm. In humans, it is crucial for fertility preservation strategies [4,5], and in domestic animals it is frequently used as a technique to accelerate the rate of genetic improvement [6]. However, the freeze–thaw process induces destabilization of plasma membrane, nuclear alterations, changes in sperm proteins and the lateral phase separation of lipid membrane components; alters the permeability of plasma membrane to water and ions; and impairs mitochondrial activity [7,8,9,10]. Proteins and lipids in cryoprotective media provide a partial protection from these deleterious effects (reviewed in [7]). Freezability or cryotolerance refers to the resilience of sperm cells to cryopreservation and shows great variability between species, mainly due to differences in membrane composition [8,11,12]. Cryotolerance is lower in pig than in bovine and human sperm; this low ability of pig sperm to withstand freeze–thaw damage is related to the low content of cholesterol [13] and high content of polyunsaturated fatty acids [14] of the plasma membrane. Moreover, great variability in sperm cryotolerance has been observed between breeds, between individuals of the same breed, and even between ejaculates from the same animal [7,10,15].

Recent research has been focused on the identification of cryotolerance markers for sperm of different species, including the pig [9,10,15]. A wide variety of proteins have been proposed as candidate biomarkers of sperm freezability, ranging from membrane transporters and receptors to enzymes [9,10,15,16,17,18,19]. Several membrane transporters exert a crucial role during sperm cryopreservation, among them to highlight hexose transporters [16,20] and aquaporins and aquaglyceroporins [9,21]. Nevertheless, few data exist about the role of ion channels during sperm cryopreservation, not only in pigs but also in other mammalian species, including humans.

While previous studies have immunolocalized both SLO1 potassium channels and voltage-gated proton channels (HVCN1) in the plasma membrane of mammalian spermatozoa (human [22,23,24], cattle [1], and pig [25,26]) and support a role for them during sperm capacitation, none has investigated whether they are involved in sperm cryotolerance. Therefore, the main objective of the present research is to elucidate the physiological role of SLO1 and HVCN1 channels during mammalian sperm cryopreservation. The study uses the pig as a model, as the presence and functional relevance of these channels during sperm capacitation have been previously reported in this species [25,26].

This functional approach was performed pharmacologically by the addition of specific SLO1- and HVCN1-blockers to cryopreservation media and further analysis of sperm quality in frozen–thawed samples. Three different blockers were used: (1) Paxilline (PAX), a fungal indole alkaloid that acts as a potent and specific inhibitor of SLO1 channels [27,28]; (2) tetraethyl ammonium chloride (TEA), a quaternary ammonium compound with a broad inhibiting effect on several K^+^ transporters [27,29]; and (3) 2-guanidino benzimidazole (2-GBI), a specific inhibitor of HVCN1 channels [30,31]. As aforementioned, all these blockers have been previously used to analyze the function of SLO1 [25] and HVCN1 channels [26] during in vitro capacitation of pig spermatozoa. 

## 2. Results

To understand the general role of K^+^ channels and the specific role SLO1 and HVCN1 channels during sperm cryopreservation, two experiments were designed. In Experiment 1, sperm samples were cryopreserved in the presence of TEA, PAX, or 2-GBI blockers which were added to LEY medium at 15 °C. In Experiment 2, blockers were added to LEYGO medium at 5 °C.

### 2.1. Sperm Viability

Post-thaw sperm viability, expressed as the percentage of spermatozoa with an intact plasma membrane, did not differ significantly between the control and samples blocked with TEA and PAX in Experiment 1 (*P* > 0.05) and Experiment 2 (*P* > 0.05). In contrast, the addition of 2-GBI to LEY medium at 15 °C (Experiment 1) and to LEYGO medium at 5 °C (Experiment 2) led to a significant decrease in sperm viability at both 30 min and 240 min post-thaw (*P* < 0.05) as compared to the control and TEA- and PAX-blocked samples (Figure 1).

### 2.2. Sperm Motility

In Experiment 1, percentages of total and progressively motile spermatozoa did not differ between the control and samples blocked with TEA and PAX at either 30- or 240-min post-thaw (*P* > 0.05; Figure 2a,c). The addition of 2-GBI resulted in a significant decrease in total motility at 30 min post-thaw (*P* < 0.05), but not in progressive motility (*P* > 0.05). After 240 min of thawing, sperm motility did not differ between the control and blocked samples (*P* > 0.05). The analysis of sperm kinematics showed a lack of significant differences between the control and samples blocked with TEA and PAX in post-thaw curvilinear (VCL) and straight-line (VSL) velocities (*P* > 0.05). In contrast, HVCN1-blocking with 2-GBI led to a significant decrease in VCL at 30 min post-thaw (*P* < 0.05) and in VSL at both 30 min and 240 min post-thaw (*P* < 0.05) (Figure 3a,c). Other kinematics parameters that were evaluated showed similar trends (data not shown).

In experiment 2, post-thaw sperm motility did not differ significantly between the control and blocked samples, regardless of the inhibitor (*P* > 0.05; Figure 2b,d). The comparative analysis of sperm kinematics showed a different effect depending on the blocker. Therefore, as compared to the control, blocking samples with TEA led to a significant decrease in VSL at 240 min post-thaw (*P* < 0.05). In addition, while samples blocked with PAX did not show changes in VCL or VSL (*P* > 0.05), inhibition of HVCN1 channels with 2-GBI led to a significant decrease in VCL at 30 min post-thaw (*P* < 0.05) and in VSL at both 30 min and 240 min post-thaw (*P* < 0.05; Figure 3b,d). Other parameters evaluated in kinematics analysis showed similar trends (data not shown).

### 2.3. Acrosome Integrity

Post-thaw acrosome integrity, expressed as percentages of viable spermatozoa with an intact acrosome, was not affected by the addition of TEA or PAX to LEY medium at 15 °C (Experiment 1; *P* > 0.05) or to LEYGO medium at 5 °C (Experiment 2; *P* > 0.05). In contrast, the addition of 2-GBI to LEY and LEYGO media led to a significant decrease of acrosome integrity at 30- and 240-min post-thawing (*P* < 0.05; Figure 4).

### 2.4. Membrane Lipid Disorder

In Experiment 1, percentages of viable spermatozoa with high membrane lipid disorder at post-thaw did not differ between the control and samples blocked with TEA and PAX (*P* > 0.05). 2-GBI blockade resulted in a significant increase in this percentage at 30 min post-thaw (*P* < 0.05). However, at 240 min post-thaw, this parameter did not differ significantly between the control and TEA- and PAX-blocked samples (*P* > 0.05; Figure 5).

In Experiment 2, the percentage of viable spermatozoa with high membrane disorder was significantly higher in 2-GBI blocked samples than in the control and samples blocked with TEA and PAX (*P* < 0.05).

### 2.5. Intracellular Levels of Calcium 

Addition of TEA and PAX blockers at either LEY medium at 15 °C (Experiment 1) or LEYGO medium at 5 °C (Experiment 2) did not alter the percentage of viable spermatozoa with high calcium levels in comparison to control samples (*P* > 0.05). In contrast, blocking HVCN1 channels with 2-GBI resulted in a significant decrease in this parameter when added to LEY and LEYGO media (*P* < 0.05; Figure 6).

### 2.6. Mitochondrial Membrane Potential

JC1_agg_/JC1_mon_ fluorescence ratio in the population of sperm with intermediate MMP (JC1_agg_^+^/JC1_mon_^+^ spermatozoa) did not differ between frozen–thawed control samples or samples blocked with PAX and TEA in either Experiment 1 (*P* > 0.05) or Experiment 2 (*P* > 0.05). In contrast, this parameter was significantly higher in 2-GBI blocked samples than in the control and samples blocked with TEA and PAX at 240 min post-thaw when added to LEYGO medium (*P* < 0.05; Figure 7b). This effect was also observed when 2-GBI was added to LEY medium, but differences were not significant (*P* > 0.05; Figure 7a).

### 2.7. Intracellular Levels of Mitochondrial Superoxide (O_2_^●−^)

In Experiment 1 and Experiment 2, the pattern of variation in the percentage of viable spermatozoa with high levels of mitochondrial superoxide did not differ between the control and samples blocked with TEA and PAX (*P* > 0.05). In contrast, the addition of 2-GBI to LEYGO medium (Experiment 2) resulted in a significant decrease in this percentage as compared to both the control and samples blocked with TEA or PAX (*P* < 0.05; Figure 8b). These differences were not significant in Experiment 1 (*P* > 0.05; Figure 8a).

### 2.8. Intracellular Levels of Hydrogen Peroxide (H_2_O_2_)

In Experiment 1, percentages of viable spermatozoa with high levels of H_2_O_2_ did not differ between the control and samples blocked with PAX and TEA at 30 min and 240 min post-thaw. In samples blocked with 2-GBI, this percentage tended to be lower than in the control and in samples blocked with TEA and PAX, but differences were not significant (*P* > 0.05; Figure 8c).

In Experiment 2, a blocking effect was observed on the percentage of viable spermatozoa with high levels of H_2_O_2_. In contrast, TEA blocker did not alter this percentage as compared to control samples at either 30 min or 240 min post-thaw (*P* > 0.05). This parameter tended to decrease in the presence of PAX blocker at 240 min post-thaw, but differences were not significant (*P* > 0.05). In contrast, in the case of the 2-GBI blocker differences became significant from 30 min post-thaw (*P* < 0.05; Figure 8d).

## 3. Discussion

Sperm cryopreservation has been extensively studied in different mammalian species, given its potential applications for safeguarding fertility. Due to the great sensitivity of sperm to freeze–thawing, previous research has been focused on both the changes in plasma membrane permeability to water and cryoprotectants [9,21] and the identification of biological markers to predict sperm cryotolerance [4,5,9,10,15,32,33,34,35]. Previous studies have also demonstrated the critical role of aquaporins (AQPs) in regulating the transport of water and cryoprotectants across cell membranes and in preventing osmotic damage [9,36,37,38]. While ion channels have been reported to be involved in sperm adaptation to changes in the osmolality of the surrounding medium [2], their physiological role during sperm cryopreservation has been less studied.

The present study analyses the general role of K^+^ channels and the specific role of SLO1 and HVCN1 channels during sperm cryopreservation. This functional approach was performed by adding distinct inhibitors (TEA, PAX, or 2-GBI) to LEY medium at 15 °C or to LEYGO medium at 5 °C; and used the pig as a model. The concentration of these three inhibitors was determined in previous studies [25,26]. The results obtained herein indicate that blocking K^+^- and HVCN1-channels during sperm cryopreservation has a different impact upon sperm quality after thawing. In effect, while TEA and PAX, which are inhibitors of K^+^ channels, did not alter post-thaw sperm function and survival as compared to the control at either 30 min or 240 min post-thaw, blocking HVCN1 channels with 2-GBI resulted in a dramatic impairment of post-thaw sperm quality.

Unexpectedly, the general blockade of K^+^ channels and the specific blockade of SLO1, either when samples were kept at 15 °C or after being cooled down to 5 °C, did not induce alterations in plasma and acrosome membrane integrity, sperm motility, and kinematics, membrane lipid disorder, intracellular calcium levels, mitochondrial membrane potential or O_2_^−^⁻ and H_2_O_2_ production. To the best of the authors’ knowledge, neither the physiological role of K^+^ channels nor that of the HVCN1 ones during sperm cryopreservation has been previously studied. On the other hand, little data exist about the nature, distribution, and function of K^+^- and HVCN1-channels in mammalian sperm. SLO channels are the main K^+^ channels in human and mouse sperm, and they are involved in the regulation of sperm volume and capacitation [39,40,41]. However, both species differ in the types and regulation mechanisms of SLO channels; in mouse sperm the predominant form is SLO3, which localizes in the principal piece [3,42,43,44,45], whereas the plasma membrane of human sperm contains SLO1 channels, which are localized in the principal piece, and SLO3 channels, which are distributed throughout the whole flagellum [40,46]. SLO1 channels are activated by both increased membrane voltage and intracellular Ca^2+^ levels [27,47], and SLO3 channels are highly sensitive to intracellular alkalinization [41]. SLO1 channels have also been identified in the acrosomal and post-acrosomal region of the sperm head, and in the mid-, principal, and terminal pieces of the flagellum of pig sperm [25]. Despite this wide distribution, some studies suggest that not only SLO1 channels but also other K^+^ channels are involved in K^+^ conductance over the pig sperm flagellum [25]. In bovine sperm, two different types of K^+^ channels have been identified, the chloride and potassium ion channel 3 (CLC-3) and the two-pore domain channel (TASK-2) [48].

Considering the physiological role of K^+^ channels during spermatogenesis, sperm maturation and capacitation [2,25], the biological significance of our present results is uncertain. Lack of differences between the control and samples blocked with TEA and PAX suggest that either K^+^ channels are not relevant for osmolar adaptation of pig sperm during cryopreservation, or that alterations in K^+^ conductance have a modest impact on post-thaw sperm quality. In this regard, further studies must be oriented towards determining the effects of cryopreservation on the relative content, functional activity, and conductance of K^+^ channels in pig sperm. Interestingly, cryopreservation of bovine sperm does not alter the amount of CLC-3 and TASK-2 channels, but it is associated to an increase of intracellular potassium levels [48]. As not only do the types of K^+^ channels differ between species but also their regulatory mechanisms, research in this field is much warranted.

K^+^ efflux is essential for sperm capacitation as it underlies the changes in motility patterns during sperm hyperactivation, plasma membrane hyperpolarization, calcium influx, and acrosome exocytosis [3,25,46,49]. The biological significance of these differences between sperm cryopreservation and capacitation remain unclear, but they seem to be related with differences in the composition of the surrounding environment. In this regard, the activation of K^+^ channels in human sperm has been strongly related with presence of divalent cations in the extracellular medium [40]. Nevertheless, further research is necessary to understand the physiological regulation of K^+^ channels in sperm cells.

The alteration of nearly all sperm parameters after 2-GBI blockade supports the physiological relevance of HVCN1 channels and of H^+^ conductance during sperm cryopreservation. HVCN1 channels have an essential regulating role in several cell types (reviewed in [50]) by driving protons unidirectionally to the extracellular medium [30], despite their functional significance in mammalian sperm being poorly understood. While HVCN1 channels are present in the plasma membrane of human, macaque, cattle and pig sperm, they are absent from mouse sperm [1,23,24,46,51]. Remarkably, in macaques, sperm cryopreservation results in a decreased content of HVCN1 channels in the plasma membrane [48].

Blockade of HVCN1 channels during cryopreservation led to reduced sperm viability, expressed as spermatozoa exhibiting an intact plasma membrane integrity, and reduced sperm motility and kinematics after thawing. In previous studies, inactivation of HVCN1 channels has been reported to result in decreased sperm viability in ejaculated human [52] and bovine sperm [1], since this makes these cells unable to regulate their inner pH [23]. Nevertheless, little data exist about the changes in the intracellular pH throughout the cryopreservation procedure. Only a single study reported a slight decrease of inner pH in cryopreserved bovine sperm [48]. Blocking HVCN1 channels also leads to a decrease in the motility and velocity of in vitro capacitated sperm from pigs [26] and humans [22]. In bovine [1] and human sperm [23], these channels are essential for activation of progressive motility upon ejaculation and for hyperactivation during capacitation. Moreover, in humans, HVCN1 activity is higher in capacitated than in non-capacitated sperm due to the phosphorylation of this channel during capacitation [22,23].

After thawing, samples blocked with 2-GBI showed reduced percentages of viable spermatozoa with high calcium levels in the sperm tail, which were determined by Fluo3 staining. These reduced intracellular calcium levels in the flagellum could be related with the low sperm motility and velocity in HVCN1-blocked samples as compared to the control. Only few data exist regarding the relationship between H^+^ and Ca^2+^ conductance. In spite of that, and in the same line of our results, Yeste et al. [26] also observed a decrease in intracellular calcium levels in the flagellum during acrosome reaction when 2-GBI was added to pig spermatozoa. Therefore, further studies are needed to elucidate the potential interaction between this inhibitor and calcium channels present in the sperm tail.

Cryopreservation has been extensively related to capacitation-like processes or cryocapacitation, which manifest in premature acrosomal exocytosis and altered plasma membrane permeability, and result in diminished sperm lifespan and the appearance of apoptotic-like changes (reviewed in [7]). As compared to control samples, blocking HVCN1 channels with 2-GBI led to a reduced percentage of viable spermatozoa with an intact acrosome and an increased percentage of viable spermatozoa with high membrane lipid disorder after thawing. In contrast, these percentages did not differ between the control and samples blocked with TEA (K^+^ channel blockade) or PAX (SLO1 channel blockade). In agreement with these findings, previous studies in pig and human sperm also demonstrated that HVCN1 blockade during in vitro capacitation induces premature acrosomal exocytosis, high plasma membrane lipid disorder and increased Ca^2+^ levels in the sperm head [26,52], whereas K^+^ channel blockade does not trigger such a sequence of events [25]. Together, these results strongly suggest that in pigs HVCN1 channels are essential for sperm homeostasis during in vitro capacitation and cryopreservation and for preventing premature sperm activation.

In control samples, cryopreservation resulted in an increase of ROS production associated to high mitochondrial membrane potential, as being extensively reported in previous studies investigating sperm cryopreservation [6,9,10]. Moreover, we also found that while blocking HVCN1 channels with 2-GBI led to increased mitochondrial membrane potential at 240 min post-thaw, it unexpectedly reduced O_2_^−^⁻ and H_2_O_2_ levels at both 30- and 240-min post-thaw. In contrast, the blockade of aquaporin channels prior to pig sperm cryopreservation results in an increase of both mitochondrial membrane potential and intracellular ROS levels [9]. This increase in intracellular ROS levels seems to be a direct consequence of the blocking of these channels, since aquaglyceroporins are not exclusively permeable to water, but also to other small molecules, such as H_2_O_2,_ which might accumulate intracellularly [9]. It must be considered that sperm cytoplasm contains a set of antioxidant enzymes, such as glutathione peroxidase (GPx), superoxide dismutase (SOD), glutathione S-transferase Mu 3 (GSTM3), and AMP-activated protein kinase (AMPK), which are damaged during cryopreservation, thus resulting in a low ability of sperm to scavenge ROS [6,10]. Nevertheless, in pig sperm, the protective effect against ROS damage of antioxidative agents added to freezing media is associated to an increased activity of antioxidant enzymes [6]. Taken all the aforementioned into account and considering that HVCN1 channels are involved in the generation of superoxide radicals from mitochondrial membrane chain by activation of NADPH oxidase [53,54], one could surmise that the decrease in ROS levels might be a direct consequence of HVCN1 blockade during sperm cryopreservation.

## 4. Materials and Methods 

### 4.1. Materials

Chemicals and fluorochromes were provided by either Sigma-Aldrich Química (Madrid, Spain) or ThermoFisher Scientific (Waltham, MA, USA). In case of another supplier, it is specified next to the product.

### 4.2. Semen samples

The present study included ejaculates from 17 Piétrain boars from an artificial insemination program, and an age range between 18 and 24 months. Boars were randomly distributed into two groups, one per experiment, of 8 and 9 males, respectively. According to the Animal Welfare Regulations of the Regional Government of Catalonia (Spain), animals were hosted in standard conditions of temperature and humidity. Feed was provided once daily and water ad libitum. Semen samples were obtained twice a week following the standard protocols for commercial seminal doses in a local farm (Semen Cardona, Cardona, Barcelona, Spain). Considering that authors did not manipulate any animal, specific approval from an Ethics Committee was not needed.

Immediately after collection, the sperm-rich fraction of each ejaculate was filtered through gauze, and diluted 1:1 (v:v) in a long-term extender (Vitasem, Magapor, S.L., Zaragoza, Spain) at 37 °C inside a collecting recipient. Diluted sperm-rich fraction was cooled to 17 °C and were transported to the lab in a thermal insulated box. Time between semen extraction and dose arrival to the lab never exceeded five hours. Before sample processing, diluted sperm-rich fraction was divided into two fractions, a fresh fraction and a fraction intended to cryopreservation. After assessing the sperm quality in the fresh fraction, the other fraction was processed for cryopreservation either in the absence or presence of inhibitors.

### 4.3. Sperm cryopreservation

Th fraction intended to cryopreservation was divided into 50-mL aliquots and centrifuged at 15 °C and 2400× *g* for 3 min. The procedure included five steps [6,7]: (1) Pellet resuspension in β-lactose-egg yolk freezing medium (LEY medium: 80% (*v*/*v*) lactose and 20% (*v*/*v*) egg yolk) at 15 °C to a final concentration of 1.5 × 10^9^ spermatozoa/mL; (2) sample dilution in LEY and cooling down to 5 °C at a ratio of − 0.1 °C/min (180 min) in a programmable freezer (Icecube 14S-B; Minitüb Ibérica SL; Tarragona, Spain); (3) sample cooling at 5 °C and dilution in LEYGO medium (LEY medium supplemented with 6% (*v*/*v*) glycerol and 1.5% Orvus ES Paste; Equex STM; Nova Chemical Sales Inc., Scituate, MA, USA) to 1 × 10^9^ spermatozoa/mL; (4) packaging of cooled and diluted samples into 0.5-mL plastic straws (Minitüb Ibérica, S.L.) and frozen in a programmable freezer (Icetube 14S-B; Minitub Ibérica, S.L), with the freezing ramp [17]: −6 °C/min from 5 °C to −5 °C (100 s), −39.82 °C/min from −5 °C to −80 °C (113 s), holding at −80 °C for 30 s, and cooling at −60 °C/min from −80 °C to −150 °C (70 s); and (5) plunging of frozen straws into liquid nitrogen (−196 °C) for long-term storage.

### 4.4. Experimental Design 

To determine the involvement of HVCN1 and SLO1 channels on sperm cryotolerance, samples were cryopreserved in the presence or the absence of either paxilline (PAX), tetraethyl ammonium chloride (TEA), or 2-guanidinobenzimidazole (2-GBI). Two sets of experiments were designed.

In the first experiment, inhibitors were added to aliquots containing sperm resuspended with LEY at 15 °C (Step 1). For this purpose, aliquots diluted with LEY were distributed into four subfractions, one for each inhibitor (i.e., PAX, TEA, and 2-GBI) and a non-treated control. In the second experiment, inhibitors were added to aliquots containing LEYGO at 5 °C (Step 3). Again, aliquots were split into four subfractions, one for each inhibitor plus a non-treated control. In both experiments, the final inhibitor concentration was 100 nM for PAX, 20 mM for TEA, and 10 mM for 2-GBI. The concentration of each inhibitor was established according to previous studies [25,26].

The analyze the effects of channel blocking during sperm cryopreservation, three straws per sample and inhibitor were thawed by drowning and agitating in a water bath at 38 °C for 15 s. Thawed samples were then diluted 1:3 (*v*/*v*) in pre-warmed Beltsville Thawing Solution (BTS) [55] and incubated at 38 °C for 240 min. Throughout incubation the sperm quality was analyzed at two critical points: 30 min and 240 min.

### 4.5. Sperm Motility

In fresh and frozen–thawed samples, the analysis of sperm motility was performed in a computer-assisted sperm analysis (CASA) system, formed by a phase contrast microscope (Olympus BX41; Olympus, Tokyo, Japan) with a video camera and an ISAS software (Integrated Sperm Analysis System V1.0; Proiser SL, Valencia, Spain). Before motility assessment, fresh samples were incubated at 38 °C for 15 min; since thawed samples were incubated at 38 °C, they were directly examined at 30 min and 240 min post-thaw. For both fresh and thawed samples, a 5 μL-volume were placed onto a pre-warmed (38 °C) Makler counting chamber (Sefi-Medical Instruments, Haifa, Israel) and observed under a negative phase-contrast field objective (Olympus 10× 0.30 PLAN). Sperm motility was assessed from the count of at least 1,000 sperm per replicate, being three replicates per sample analyzed.

For each sperm sample and inhibitor, different motility and kinematics parameters were evaluated: Total (TMOT, %), progressive sperm motility (PMOT, %) curvilinear velocity (VCL, μm/s); straight line velocity (VSL, μm/s); average path velocity (VAP, μm/s); amplitude of lateral head displacement (ALH, μm); beat cross frequency (BCF, Hz); linearity (LIN=VSL/VCL × 100, %); and straightness (STR = VSL/VAP × 100, %); wobble (WOB:VAP/VCL × 100, %). Only those spermatozoa with VAP equal to or higher than 10 μm/s were considered motile, and those with STR equal to or higher than 45% were considered as progressively motile. Each motility and kinematic sperm parameter was expressed as the mean ± standard error of the mean (SEM).

### 4.6. Flow Cytometry

Flow cytometry was used to assess the plasma and acrosome membrane integrity, membrane lipid disorder, intracellular calcium levels, mitochondrial membrane potential (MMP), intracellular levels of superoxide (O_2_^●−^) radicals and intracellular levels of hydrogen peroxide (H_2_O_2_). Each assay required pre-dilution of both fresh and frozen–thawed samples to a final concentration of 1×10^6^ spermatozoa/mL, as well as an appropriate fluorochrome combination and further incubation at 38 °C in darkness. For each sample and sperm parameter analyses were performed per triplicate.

All cytometric analyses were performed in a Cell Laboratory QuantaSC™ cytometer (Beckman Coulter; Fullerton, CA, USA), using an argon ion laser (488 nm) at 22 mW. Cell diameter/volume was assessed using the Coulter principle for volume assessment of the cytometer, which measured the changes in electrical resistance produced in an electrolyte solution by non-conductive particles. In this system, forward scatter (FS) is replaced by electronic volume (EV). EV-channel calibration was performed using 10-μm Flow-Check fluorospheres (Beckman Coulter), by positioning this size of bead at channel 200 on the EV-scale.

Three optical filters were used: FL1 filter (Dichroic/Splitter, DRLP: 550 nm, BP filter: 525 nm, detection width: 505–545 nm) to detect green emission from SYBR-14, Fluo3, PNA-FITC, YO-PRO-1, JC-1 monomers (JC-1_mon_) and 2′,7′-dichlorofluorescein (DCF^+^) fluorochromes; FL2 filter (DRLP: 600 nm, BP filter: 575 nm, detection width: 560–590 nm) to detect orange emission from JC-1 aggregates (JC-1_agg_); and FL3 (LP filter: 670 nm/730 nm, detection width: 655–685 nm), to detect red emission from propidium iodide (PI), merocyanine 540 (M540) and Mito-ethidium (Mito-E^+^). The signal was logarithmically amplified, and the adjustment of photomultiplier settings was performed according to the staining method.

EV and side scatter (SS) were measured and linearly recorded for all particles. The sheath flow rate was of 4.17 μL/min, and the number of events counted per replicate of 10,000. The analyzer threshold of the EV channel was adjusted to exclude both subcellular debris and cell aggregates, with a diameter < 7 μm and > 12 μm, respectively. According to EV and SS distributions, only sperm-specific events were positively gated, whereas other events were gated out.

Cytometric data analysis was performed using the Flowing Software (Ver. 2.5.1; University of Turku, Finland), following the recommendations of the International Society for Advancement of Cytometry (ISAC). For each parameter, the results are expressed as the mean ± SEM.

#### 4.6.1. Plasma and Acrosome Membrane Integrity

Plasma membrane integrity was used as an indicator of sperm viability. The assay was performed with the LIVE/DEAD Sperm Viability Kit (Molecular Probes, Eugene, OR, USA), which consisted in two consecutive staining steps [56]: (1) Incubation of sperm samples with SYBR-14 (final concentration: 100 nmol/L) for 10 min, and (2) incubation with PI (final concentration of 12 μmol/L) for 5 min. Two different sperm populations were differentiated according to the staining pattern: (1) Viable sperm, with a staining pattern of SYBR-14^+^/PI^−^ and green fluorescence emission, and (2) non-viable sperm, which could stain either SYBR-14^−^/PI^+^ and SYBR-14^+^/PI^+^, being the fluorescence emission red or red and green, respectively. A population of unstained (SYBR-14^−^/PI^−^) particles was also observed; this population corresponded to non-sperm particles and it was used to correct the proportions of intact sperm found in the other tests [57]. Results are expressed as the percentage of viable spermatozoa (SYBR-14^+^/PI^−^) (mean ± SEM; *n* = 8 in the first experiment and *n* = 9 in the second experiment).

Acrosome membrane integrity was assessed by the double-staining with PNA-FITC and PI [58], both added simultaneously to the sperm samples. The final concentration was of 2.5 µg/mL for PNA-FITC and of 12 µmol/L for PI, and the incubation time of 10 min in dark. This procedure let to classify the sperm cells into two categories: (1) spermatozoa with an intact plasma membrane and acrosome (PNA-FITC^−^/PI^−^); and (2) spermatozoa with a damaged acrosome and/or plasma membrane damage. This second category included three different sperm populations with a specific staining pattern: PNA-FITC^+^/PI^−^ pattern, which corresponded to those spermatozoa with a damaged plasma membrane, PNA-FITC^+^/PI^+^ pattern, which included those spermatozoa with a damaged plasma membrane and a partially altered outer acrosome membrane, and PNA-FITC^−^/PI^+^ pattern, from those spermatozoa with a damaged plasma membrane and a lost outer acrosome membrane. Events with PNA-FITC^−^/PI^−^ pattern were corrected using the percentages of non-sperm debris particles (SYBR14^−^/PI^−^) obtained from the viability assay. Only the percentage of sperm with PNA-FITC^−^/PI^−^ staining pattern were showed (mean ± SEM; *n* = 8 in the first experiment and *n* = 9 in the second experiment).

#### 4.6.2. Plasma Membrane Lipid Disorder

Plasma membrane lipid disorder was evaluated using the fluorochrome M540, which can detect alterations in packing order of phospholipids in the external leaflet [59,60]. Sperm samples were double incubated with M540 (final concentration: 2.6 μmol/L) and YO-PRO-1 (final concentration: 25 nmol/L) for 10 min in darkness. Flow cytometry led to identify viable (YO-PRO-1^−^) and non-viable (YO-PRO-1^+^) sperm populations showing either low (M540^−^) or high (M540^+^) membrane lipid disorder. As in other sperm parameters, the percentage of viable sperm with low membrane lipid disorder (M540^−^/YO-PRO-1^−^) was corrected using the non-sperm particles from the SYBR-14/PI staining. Results are expressed as the percentage of viable sperm with high membrane lipid disorder (M540^+^/YO-PRO-1^−^) (mean ± SEM; *n* = 8 in the first experiment and *n* = 9 in the second experiment).

#### 4.6.3. Determination of Intracellular Calcium Levels

Intracellular calcium levels were evaluated using the fluorochrome Fluo3-acetomethoxy ester (Fluo3-AM, F-1241; Molecular Probes, Invitrogen, ThermoFisher Scientific) and PI. Both fluorochromes were added together to the sperm samples at a final concentration of 1 µmol/L and 12 µmol/L, respectively, and incubated for 10 min in darkness [61]. Four different sperm populations with a specific staining pattern were distinguished by flow cytometry: (1) Viable spermatozoa with low calcium levels (Fluo3⁻/PI⁻); (2) viable spermatozoa with high calcium levels (Fluo3^+^/PI⁻); (3) non-viable spermatozoa with low calcium levels (Fluo3⁻/PI^+^); and (4) non-viable spermatozoa with high calcium levels (Fluo3^+^/PI^+^). FL1 spill over into the FL3-channel (2.45%) and FL3 spill over into the FL1-channel (28.72%) were compensated. Percentages of debris particles (SYBR14^-^/PI^-^) were used to correct the percentages of Fluo3^-^/PI^-^ spermatozoa, and to recalculate the percentages of the rest of sperm populations. For each sperm sample and cryopreservation step, only the percentage of viable spermatozoa with high intracellular calcium levels were recorded (Fluo3^+^/PI^-^) (mean ± SEM; n = 8 in the first experiment and *n* = 9 in the second experiment).

#### 4.6.4. Mitochondrial Membrane Potential (MMP)

To assess mitochondrial membrane potential (MMP), sperm samples were stained for 30 min with the fluorochrome JC-1 at a final concentration of 0.3 μmol/L [62]. The principle of this staining procedure is that the increase in MMP results in JC-1 aggregation (JC-1_agg_), whereas low MMP maintains JC-1 in its monomeric form (JC-1_mon_). Flow cytometry dot plots provided three different sperm populations: (1) Sperm with low MMP (JC-1_mon_), producing green fluorescence; (2) sperm with high MMP (JC-1_agg_), producing orange fluorescence; and (3) sperm with intermediate MMP (JC-1_mon_ and JC-1_agg_), producing both green and orange fluorescence. Particles with no fluorescence (either green or orange) were gated out. Data were compensated, by subtracting green fluorescence from FL2-channel (51.70%). Results are expressed as the ratio between JC1_agg_/JC1_mon_ sperm populations (mean ± SEM; *n* = 8 in the first experiment and *n* = 9 in the second experiment).

#### 4.6.5. Intracellular Levels of Mitochondrial Superoxide (O_2_^−^⁻)

Intracellular levels of mitochondrial superoxide (O_2_^−^⁻) radicals were determined from the double-staining with MitoSOX and YO-PRO-1 [63]. MitoSOX or Mito-Hydroethidine is a positively charged fluorochrome that accumulates in mitochondria [64], being oxidized by O_2_^−^⁻ radicals to a fluorescent molecule (Mito-ethidium, Mito-E), which is positively charged (Mito-E^+^). The staining pattern consisted in the double-incubation of sperm samples with MitoSOX (final concentration: 4 μmol/L) and YO-PRO-1 (final concentration: 40 nmol/L) for 20 min in darkness. The subsequent analysis by flow cytometry let to differentiate viable (YO-PRO-1^−^) and non-viable (YO-PRO-1^−^) sperm cells with either low (Mito-E^−^) or high (Mito-E^+^) mitochondrial superoxide levels. The percentage of viable sperm with low mitochondrial superoxide levels (Mito-E^−^/YO-PRO-1^−^) was corrected using the non-sperm particles (SYBR-14^-^/PI^-^), and the percentages the other three sperm populations were recalculated. Results are expressed as the percentage of viable sperm with high levels of mitochondrial superoxide (Mito-E^+^/YO-PRO-1^−^) (mean ± SEM; *n* = 8 in the first experiment and *n* = 9 in the second experiment).

#### 4.6.6. Intracellular Levels of Hydrogen Peroxide (H_2_O_2_)

Intracellular levels of hydrogen peroxide (H_2_O_2_) were analyzed using 2′,7′-dichlorodihydrofluorescein diacetate (H_2_DCFDA), which can penetrate the sperm cell membrane, being then converted into a highly fluorescent molecule, 2′,7′-dichlorofluorescein (DCF^+^) by oxidation. According to the protocol of Guthrie and Welch [63], sperm samples were incubated with H_2_DCFDA and PI for 30 min, at final concentration of 200 μmol/L and 12 μmol/L, respectively. This procedure led to differentiate viable (PI^−^) and non-viable sperm populations (PI^−^), with either low (DCF^−^) or high (DCF^+^) peroxide levels. The percentage of viable sperm with high peroxide levels (DCF^+^/PI^−^) was corrected as previously indicated. Results are expressed as the percentage of viable sperm with low (DCF^−^/PI^−^) and high (DCF^+^/PI^−^) peroxide levels (mean ± SEM; *n* = 8 in the first experiment and *n* = 9 in the second experiment).

### 4.7. Statistical Analyses

Statistical analyses of data were performed with the package IBM SPSS Statistics 25.0 (Armonk, New York, NY, USA), using a mixed model for each one of the sperm parameters measured. In this mixed model, the cryopreservation steps (i.e., fresh and frozen–thawed at 30 min and 240 min) were considered as intra-subject factors, whereas the inhibitors (i.e., control, PAX, TEA and 2-GBI) were considered as fixed-effects factors. A post-hoc Sidak test was also used for pair-wise comparisons. Before the mixed model and post-hoc analyses, both Shapiro–Wilk and Levene tests were run to check the distribution of data and homogeneity of variances. In all statistical analyses the significance level was of *P* ≤ 0.05. All the results are expressed as the mean ± SEM.

## 5. Conclusions

In conclusion, inhibition of K^+^ channels and SLO1 channels does not affect the sperm ability to withstand cryopreservation. In contrast, blocking HVCN1 channels during freezing severely impairs sperm function and survival after thawing, thus indicating that they are relevant for mammalian sperm cryotolerance. Interestingly, HVCN1-blockade was found to induce a decrease in the percentages of viable spermatozoa with high ROS levels. These remarkable findings warrant further research, since they may extend our understanding on the physiological changes associated to sperm cryotolerance and may help develop new strategies to improve cryopreservation protocols. 

## Figures and Tables

**Figure 1 ijms-22-01646-f001:**
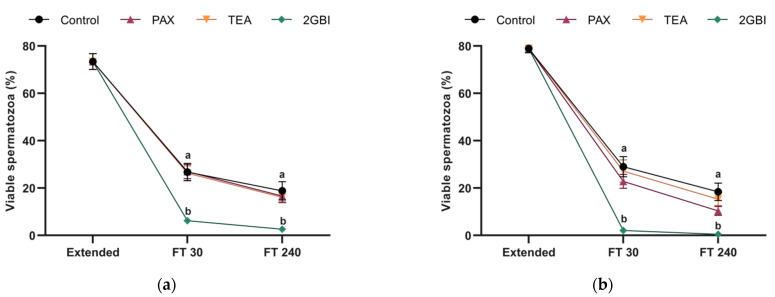
Percentages of viable spermatozoa (SYBR14^+^/PI^−^) in extended and frozen–thawed (FT) samples at 30 min and 240 min post-thaw. In Experiment 1 (*n* = 8), TEA, PAX, and 2-GBI blockers were added to LEY medium at 15 °C (**a**), whereas in Experiment 2 (*n* = 9), they were added to LEYGO medium at 5 °C (**b**). Different superscripts indicate significant differences (*P* < 0.05) between samples within the same time point. Results are given as mean ± SEM.

**Figure 2 ijms-22-01646-f002:**
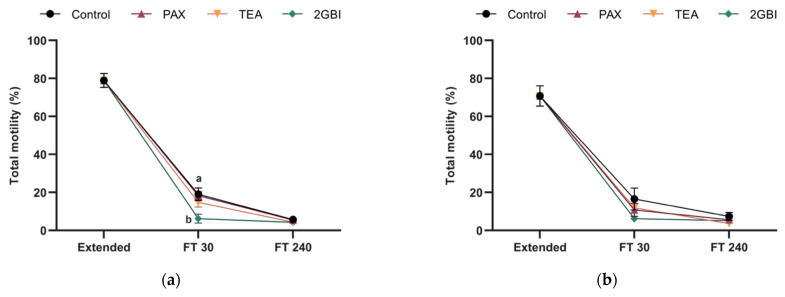
Percentages of total (**a**,**b**) and progressive (**c**,**d**) motile spermatozoa in extended and frozen–thawed (FT) samples at 30 min and 240 min post-thaw. In Experiment 1 (*n* = 8), TEA, PAX, and 2-GBI blockers were added to LEY medium at 15 °C (**a**,**c**), whereas in Experiment 2 (*n* = 9), they were added to LEYGO medium at 5 °C (**b**,**d**). Different superscripts indicate significant differences (*P* < 0.05) between samples within the same time point. Results are given as mean ± SEM.

**Figure 3 ijms-22-01646-f003:**
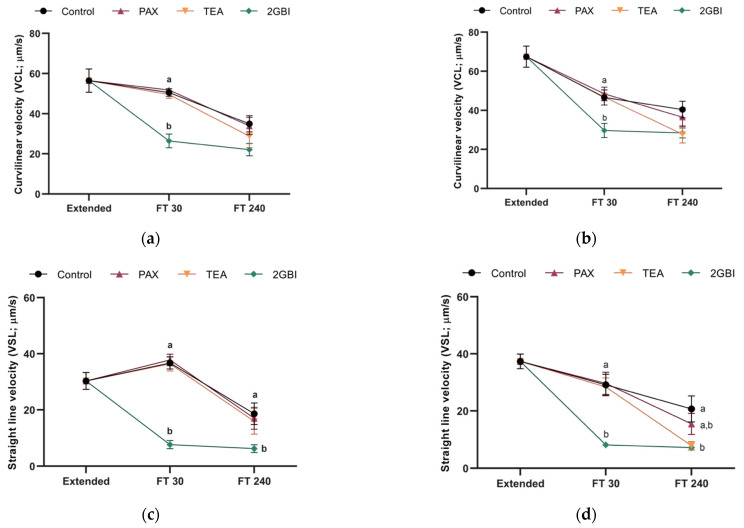
Curvilinear velocity (VCL; **a**,**b**) and straight-line velocity (VSL; **c**,**d**) in extended and frozen–thawed (FT) samples at 30 min and 240 min post-thaw. In Experiment 1 (*n* = 8), TEA, PAX, and 2-GBI blockers were added to LEY medium at 15 °C (**a**,**c**), whereas in Experiment 2 (*n* = 9), they were added to LEYGO medium at 5 °C (**b**,**d**). Different superscripts indicate significant differences (*P* < 0.05) between samples within the same time point. Results are given as mean ± SEM.

**Figure 4 ijms-22-01646-f004:**
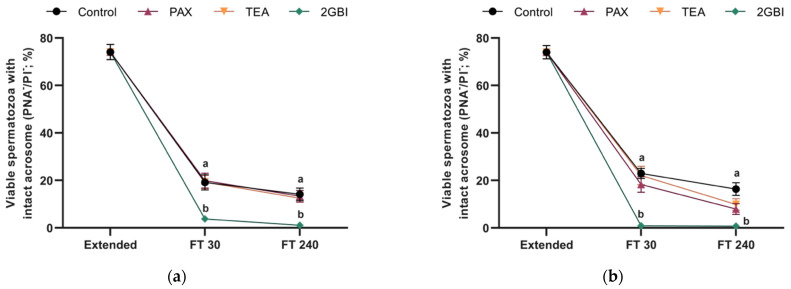
Percentages of viable spermatozoa with an intact acrosome in extended and frozen-thawed (FT) samples at 30 min and 240 min post-thaw. In experiment 1 (*n* = 8), TEA, PAX, and 2-GBI blockers were added to LEY medium at 15 °C (**a**), whereas in experiment 2 (*n* = 9), they were added to LEYGO medium at 5 °C (**b**). Different superscripts indicate significant differences (*P* < 0.05) between samples within the same time point. Results are given as mean ± SEM.

**Figure 5 ijms-22-01646-f005:**
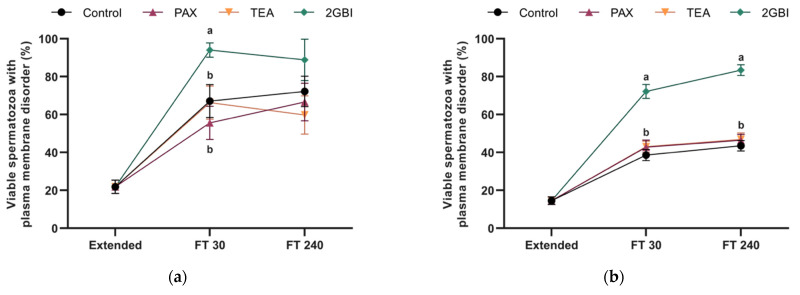
Percentages of viable spermatozoa with high membrane lipid disorder in extended and frozen–thawed (FT) samples at 30 min and 240 min post-thaw. In Experiment 1 (*n* = 8), TEA, PAX, and 2-GBI blockers were added to LEY medium at 15 °C (**a**), whereas in Experiment 2 (*n* = 9), they were added to LEYGO medium at 5 °C (**b**). Different superscripts indicate significant differences (*P* < 0.05) between samples within the same time point. Results are given as mean ± SEM.

**Figure 6 ijms-22-01646-f006:**
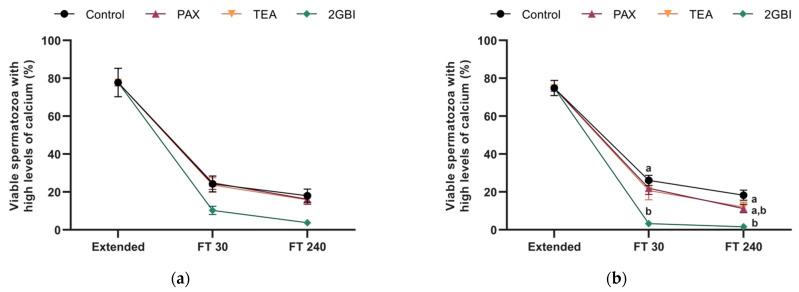
Percentages of viable spermatozoa with high levels of intracellular calcium in extended samples and frozen—30 min and 240 min post-thaw. In Experiment 1 (*n* = 8), TEA, PAX, and 2-GBI blockers were added to LEY medium at 15 °C (**a**), whereas in Experiment 2 (*n* = 9), they were added to LEYGO medium at 5 °C (**b**). Different superscripts indicate significant differences (*P* < 0.05) between samples within the same time point. Results are given as mean ± SEM.

**Figure 7 ijms-22-01646-f007:**
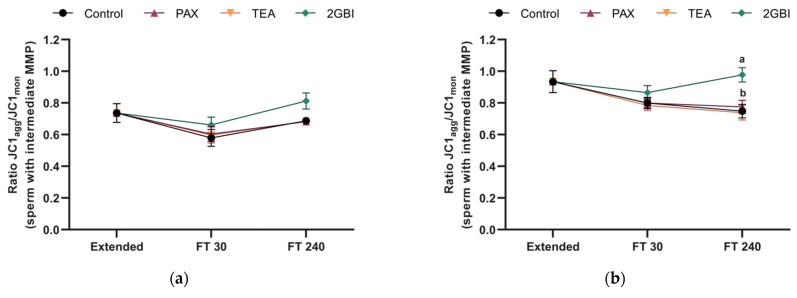
JC1_agg_/JC1_mon_ ratios in extended and frozen–thawed (FT) samples at 30 min and 240 min post-thaw. In Experiment 1 (*n* = 8), TEA, PAX, and 2-GBI blockers were added to LEY medium at 15 °C (**a**), whereas in Experiment 2 (*n* = 9), they were added to LEYGO medium at 5 °C (**b**). Different superscripts indicate significant differences (*P* < 0.05) between samples within the same time point. Results are given as mean ± SEM.

**Figure 8 ijms-22-01646-f008:**
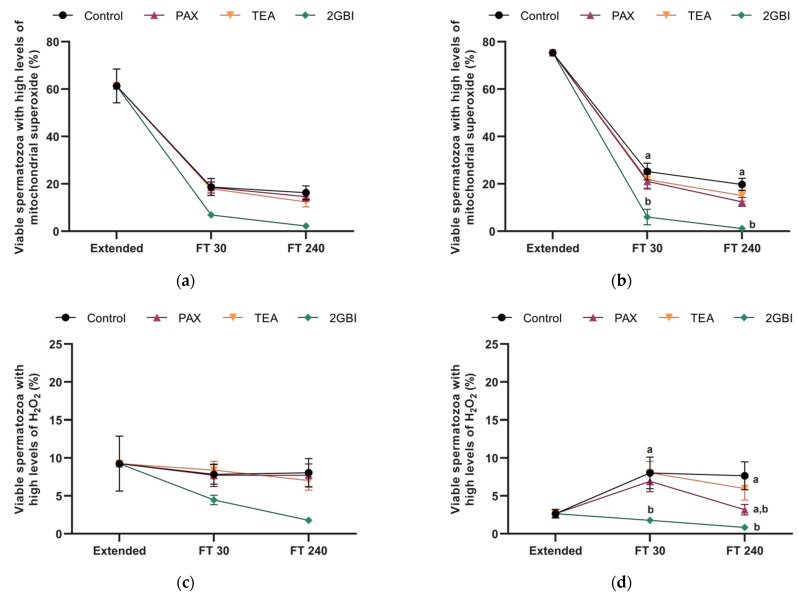
Mitochondrial superoxide levels expressed as percentages of viable spermatozoa Mito-E^+^/YO-PRO-1⁻ (**a**,**b**), and peroxide levels expressed as percentages of viable spermatozoa DCF^+^/PI⁻, (**c**,**d**) in extended and frozen–thawed (FT) samples at 30 min and 240 min post-thaw. In Experiment 1 (*n* = 8), TEA, PAX, and 2-GBI blockers were added to LEY medium at 15 °C (**a**,**c**), whereas in Experiment 2 (*n* = 9), they were added to LEYGO medium at 5 °C (**b**,**d**). Different superscripts indicate significant differences (*P* < 0.05) between samples within the same time point. Results are given as mean ± SEM.

## Data Availability

The data presented in this study are available on request from the corresponding author.

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
