# Peer review of "HVCN1 but Not Potassium Channels Are Related to Mammalian Sperm Cryotolerance"

_ijms, 2021, doi:10.3390/ijms22041646_

Round 1

Reviewer 1 Report

8th January, 2021

Review of Manuscript ID: ijms-1077618, by Delgado-Bermudez A. et al., entitled: “HVCN1 but not potassium channels are related to mammalian  sperm cryotolerance” that is intended for publication in International Journal of Molecular Sciences.

The effectiveness of porcine semen freezing is still unsatisfactory due to its high susceptibility to sperm damage during cryopreservation.

The Authors have undertaken research to determine, for the first time, the importance of SLO1 potassium channels and voltage-gated proton channels (HVCN1) for cryotolerance processes during freezing/thawing the spermatozoa of mammalian species, including pigs. For this reason, they used specific channel inhibitors such as TEA and PAX or 2-GBI that are served as supplements of cryoprotective media. The impact of these selective inhibitors on a variety of parameters related to boar semen quality was investigated. These semen quality parameters encompass: 1) sperm viability; 2) sperm motility; 3) acrosome integrity; 4) plasma membrane lipid disorder; 5) intracellular levels of calcium cations; and 6) intracellular levels of mitochondrial superoxide and hydrogen peroxide.

 It is worth emphasizing that a rich methodological workshop with the use of modern techniques of spermatological diagnostics such as computer-assisted sperm analysis (CASA) system, flow cytometry, fluorocytochemistry and fluorescence microscopy has been used to assess the involvement of these channels in boar sperm crytolerance. The Authors have obtained valuable results presented in the form of many charts. They have shown that inhibition of SLO1 potassium channels mediated by two blockers (TEA or PAX) does not affect the ability of boar spermatozoa to withstand the cryopreservation. Nonetheless, 2-GBI-dependent inhibition of  HVCN1 channels during freezing significantly impairs the function of spermatozoa and their survival rate after thawing, which is reflected in a decrease of the semen quality-related parameters evaluated.

Furthermore, it is noteworthy to mention that the Authors have used the relevant methods for statistical analyzing the results and have selected adequate references. This enabled to comprehensively interpret and critically evaluate the results obtained by the Authors as compared to the results achieved by other investigators. Generally, the paper is interesting and very well written in English. The manuscript has been prepared in the format compatible with the requirements of International Journal of Molecular Sciences.

In conclusion, I recommend the Editorial Board of International Journal of Molecular Sciences to allow for publication of this manuscript in its present form.

Author Response

We sincerely appreciate your positive assessment and recommendation.

Reviewer 2 Report

The title generalizes the work to all mammals, but it has only been done with pigs. You must modify this aspect.

The first part of the discussion is very long. It must be summarized.

And further discuss the results obtained in the study carried out.

Correct design, and good presentation of results.

Author Response

Thank you very much for your comments.

1) The title generalizes the work to all mammals, but it has only been done with pigs. You must modify this aspect.

Please, note that there are not previous literature on the physiological role of ion channels in sperm cryopreservation, so we consider this study as the basis for futher research in mammals.

2) The first part of the discussion is very long. It must be summarized.

In the original version we did a huge effort to summarize this section. Please, note that this manuscript includes two different topics, ion channels and cryopreservation, and alltogether made the Discussion a little long.

3) And further discuss the results obtained in the study carried out.

We have carefully revised the manuscript and we consider that the description of the results obtained is appropriate.